# Complete Mitochondrial Genome Characterization and Phylogenomics of the Stingless Bee, *Heterotrigona itama* (Apidae: Meliponini)

**DOI:** 10.3390/insects16050535

**Published:** 2025-05-19

**Authors:** Orawan Duangphakdee, Pisit Poolprasert, Atsalek Rattanawannee

**Affiliations:** 1Native Honeybee and Pollinator Research Center, Ratchaburi Campus, King Mongkut’s University of Technology Thonburi, Thung Khru, Bangkok 10140, Thailand; orawan.dua@kmutt.ac.th; 2Department of Entomology, Faculty of Agriculture, Kasetsart University, 50 Ngam Wong Wan Road, Lat Yao, Chatuchak, Bangkok 10900, Thailand; fagrpspo@ku.ac.th; 3Research and Lifelong Learning Center for Urban and Environmental Entomology, Kasetsart University Institute for Advanced Studies, Kasetsart University, Bangkok 10900, Thailand

**Keywords:** stingless bee, *Heterotrigona itama*, mitochondrial genome, phylogeny, systematics

## Abstract

Stingless bees serve an important part in tropical ecosystems by pollinating and producing medicinal honey. *Heterotrigona itama* is a commercially significant stingless bee species in Southeast Asia. However, overexploitation has raised questions about its conservation status in some areas, notably Thailand. We examined *H. itama*’s entire mitochondrial genome to gain a better understanding of its genetic traits and evolutionary ties. Our findings shed light on gene organization, nucleotide composition, codon usage, and evolutionary relationships among other corbiculate bees. This knowledge will help to further taxonomy, population genetics, and stingless bee conservation efforts.

## 1. Introduction

The highly eusocial stingless bees are among the most diverse corbiculate bees, classified under the tribe Meliponini of the Apinae subfamily. At least 600 species have been described and are distributed globally between the Tropic of Cancer and the Tropic of Capricorn [1,2]. The beneficial traits of stingless bee pollinators include their loyalty and constancy to specific flowers, ability to sustain perennial populations and colonies, harmless stings, ease of beekeeping, and strong worker recruitment behavior [3,4,5]. Additionally, stingless bee honey production is increasing because of its rich medicinal value [6,7,8]. Meliponine bees play essential roles in tropical ecosystems and produce valuable hives. They are also an essential example in understanding the evolution, systems, and diversity of social insects [9].

In Thailand, *Heterotrigona itama*, Cockerell, 1918 is one of the most important extensively used species in meliponiculture. This species is distributed throughout Thailand, Malaysia, Singapore, and Indonesia [10]. Meliponiculture of *H. itama* includes propagation in apiaries to sell colonies, produce honey products, and pollinate crops [11,12]. Due to the rising demand for stingless bee honey, interest in meliponiculture has also increased. Therefore, beekeeping *H. itama* can potentially increase household income in many rural areas throughout Thailand. The excessive exploitation of wild stingless bees for commercial purposes has reduced their population size, resulting in *H. itama* being allocated an at risk conservation status in Thailand.

Owing to their small size and diversity in terms of external morphology [13], as well as overlapping identification traits among species, many stingless bee species have failed to be identified, leaving the systematic and evolutionary histories of many species uncertain [14]. These factors influence thorough research and the efficient utilization of stingless bee resources. Some previous studies have presented important references for the positions of high-level taxa, whereas sufficient taxonomic data remain rare for some smaller genera, including *Heterotrigona* [12,15,16,17].

Although morphology-based classification and identification are critical, researchers typically require considerable practice to precisely identify the subtle differences among some stingless bee species [18]. On the other hand, DNA data are less reliant on expertise and can be effectively used to identify species-level taxa and clarify conflicting evolutionary histories [19]. High mutation rate and non-recombination, mitochondrial genome data are most valuable for identifying and classifying cryptic stingless bee taxa [14]. Due to their advantages as a biological marker and easy availability, mitochondrial genome data have been extensively used in identification of cryptic species, population genetics studies, evaluation of microevolutionary processes, molecular phylogenetic and systematics investigations, and genome characterization analysis [14,20].

Several previous studies indicated that complete mitochondrial genome (mitogenome) sequences are particularly effective for examining species biodiversity and conservation [18,21,22]. The molecule of bee mitogenome is a circular double-stranded one that contains 13 protein-coding genes (PCGs), 22 transfer RNAs (tRNAs), and 2 ribosomal RNAs (rRNAs) [23,24]. Additionally, the mitochondrial genome also consists of the noncoding and D-loop regions, which controls both replication and transcription of the whole mitogenome [18,25].

The objectives of this study were (1) to analyze the whole mitogenome of *H. itama*; (2) examine the structural features, base composition, bias of codon usage, and tRNA gene alteration; and (3) construct a phylogenetic tree with other hymenopterans of the family Apidae, gene order, and gene arrangement. These mitochondrial genome sequences will be useful for studying the systematic analysis and evolutionary history of this species, thereby providing insights for taxonomic and conservation biology.

## 2. Materials and Methods

### 2.1. Sample Collection and Species Identification

The adult worker specimens of *H. itama* were obtained from managed colonies belonging to two commercial meliponaries located in the Narathiwat (three samples: 6°24′11″ N; 101°42′22″ E) and Nakhon Sri Thammarat (two samples: 8°05′09″ N; 99°52′46″ E) provinces. Bee samples were collected from the entrance of the hive. All bee samples were initially preserved in absolute ethanol, kept at −20 °C, and deposited in the insect specimen room of the Department of Entomology, Faculty of Agriculture, Kasetsart University (ENTO‒AGR‒KU), Thailand, and the specimen’s voucher was archived (ENTOKU Voucher HI01‒HI05). Species confirmation was carried out based on the taxonomic literature guidelines provided by Samsudin et al. [16] and Trianto et al. [17]. Species identification was confirmed by DNA barcoding at the Ento‒AGR‒KU.

### 2.2. Ethics Statement

In this study, written consent was obtained from stingless beekeepers, who approved and agreed to participate in the research by allowing bee sample collection from their farms. Farm owners assisted in preparing colonies for sample collection. No special permits were required, as the study did not involve endangered or protected species. The number of stingless bee samples collected was minimal, and ethical research standards were followed. All animal experiments adhered to the guidelines set by the Animal Experiment Committee of Kasetsart University (approval no. ACKU66−AGR−015).

### 2.3. DNA Extraction

The thorax of a single worker of *H. itama* per colony was used to extract genomic DNA using a DNeasy Blood and Tissue Kit (Qiagen, Germantown, MD, USA). The quality of extracted genomic DNA was quantified using 0.8% agarose gel electrophoresis. The quantity of the DNA was confirmed by spectrophotometric measurements at 260 and 280 nm using a Tecan Microplate Reader Infinite 200 Pro (Thermo Fisher Scientific, Zurich, Switzerland). Then, a REPLI-g mitochondrial DNA kit (Qiagen, Germantown, MD, USA) was used to isolate mitochondrial DNA from the genomic DNA according to the production manual.

### 2.4. Sequencing, Assembly, Annotation, and Analysis

In this study, the Illumina NextSeq 500 platform was used to sequence the mitogenome library of *H. itama*. Primary FASTQ files were generated. Then, the total number of nitrogenous bases, number of reads, and quality score of 30 (Q30) were determined. The fastp software (version 0.12.4) [26] was utilized to remove the adapter, junk sequences, and ambiguous reads from the high-quality data. The filtered sequences were then aligned with reference sequences using BWA software (version 0.7.17) [27]. Consensus sequences were extracted and analyzed for protein-coding genes (PCGs), transfer RNAs (tRNAs), and ribosomal RNAs (rRNAs) using SAM software (http://samtools.sourceforge.net) [28]. Finally, gene prediction for the invertebrate mitochondrial genome was performed using MITOS web server (http://mitos.bioinf.uni-leipzig.de/index.py) [29].

To construct larger stretches of DNA contigs, de novo Bruijn graph-based assembly was performed using short-read data files. The resulting contigs were utilized for annotation of the mitogenome. The Megahit assembler software (version 1.2.9) [30] implemented in the MitoZ package [31] was used to generate the mitogenomes. The quality of the genome assembly was quantified using QUAST [32]. The whole mitochondrial genome of *H. itama* was further annotated and subsequently substantiated with three mitogenome sequences, including *Tetragonula iridipenis* (OQ139639), *Tetragonula pagdeni* (OK336459.1), and *Lepidotrigona flavibasis* (MN747147), obtained from the GenBank. The secondary structure of tRNA genes of the *H. itama* mitogenome was also predicted using MITOS. Then, the GCview [33] was performed to examine the entire mitogenome map, GC content, and GC skewness.

To examine the relative synonymous codon usage (RSCU), codon usage, and the A+T bias of the 13 PCGs, the MEGA X software (version 10.2.6) [34] was performed. Then, the AT and GC skews were examined following the report of Perna and Kocher [35]. The intergenic spacers and overlapping regions between the genes of *H. itama* mitogenome were manually calculated. Finally, the complete sequences of mitochondrial genome of *H. itama* were deposited in the NCBI database (GenBank accession no: PQ759010 and PQ766639).

### 2.5. Phylogenomic Reconstruction

The taxonomic and phylogenetic relationships of *H. itama* with other corbiculate bees (Meliponini, Apini, and Bombini) were examined using 19 complete mitochondrial genome sequences obtained from GenBank.

To estimate the best-fit model for phylogeny construction of each partitioning scheme, the software Partition Finder [36] was used with a corrected Akaike Information Criterion (AICc). The GTR + I model was selected using AIC for nucleotide alignment analysis. All complete mitogenome sequences from 19 different corbiculate bee species, along with an outgroup comprising *Colletes gigas* (KM978210), *Euaspis polynesia* (MT909816.1), and *Megachile sculpturalis* (KT223644), were aligned using MEGA X software. Maximum likelihood (ML) and Bayesian inference (BI) methods were used to determine phylogenetic relationship using the online CIPRES Science Gateway [37]. The ML analysis was conducted using IQ-TREE [38], followed by 1000 ultrafast bootstrap replicates [39] to assess the bootstrap support (BS) values of the tree topology. The BI analysis was executed with MrBayes (version 3.2.7) software [40]. A clade in the reconstructed phylogenetic tree was considered strongly supported if the ultrafast bootstrap support was at least 95% and the Bayesian bipartition posterior probability was 0.95 or higher [39,41].

### 2.6. Gene Rearrangement Assessment

The complete mitochondrial genome of *H. itama*, along with those of 22 related bee species downloaded from GenBank, was analyzed for gene order and potential gene rearrangements using PhyloSuite software (http://phylosuite.jushengwu.com/) [42]. The results were then visualized on the iTOL web server [43]. To quantitatively compare the rearrangements in each mitogenome, the rearrangement score (RS) and the rearrangement frequency (RF) of each individual gene were computed using qMGR software (http://qmgr.hnnu.edu.cn/) [44].

## 3. Results

### 3.1. Organization and Nucleotide Composition

The complete circular mitochondrial genome of the Thai commercial stingless bee *H. itama* was 15,318 bp (Figure 1). Thirty-four sequence components were identified, including 22 tRNA genes, 2 rRNA genes, and 13 PCGs (Table 1 and Figure 1). The strand localization demonstrated that the H- (+) and L-strands (−) contained 28 and 9 genes, respectively (Table 1). The maximum and minimum lengths of the mitochondrion genome of *H. itama* were 1653 bp (*nad5*) and 57 bp (*tRNA-Ser*), respectively. The nitrogenous base composition in the *H. itama* mitogenome sequence was found to be 35.06% adenine (A), 42.42% thymine (T), 11.30% guanine (G), and 12.22% cytosine (C) (Table 2). The A+T bias varied between 65.22% and 85.24% across the 13 protein-coding genes (PCGs) and was 75.41% for the entire mitogenome. The AT skew ranged from −0.214 to −0.014 across the thirteen protein-coding genes (PCGs), showing a relatively stronger bias compared to the two rRNA genes, *rrnS* (0.070) and *rrnL* (0.014) (Table 2).

### 3.2. PCGs and Codon Usage Bias

Of the 15,318 bp in the complete mitogenome sequence of *H. itama*, 13 PCGs comprising 11,045 bp (72.10%) were identified. These PCGs varied in length from 168 bp (*atp8*) to 1653 bp (*nad5*). Only the nad1 gene was found on the L-strand (−), whereas the other 12 PCGs were positioned on the H-strand (+). Five start codons (ATT, ATA, ATC, ATG, and TTG) were identified in the mitogenome of *H. itama*. The ATT codon was used by six PCGs (*cox1*, *cox2*, *atp8*, *nad4l*, *nad5*, and *nad1*) as their start codon, ATG was adopted by three genes (*cox3*, *nad4*, and *cytb*), ATA was used by two genes (*nad3* and *nad6*), and ATC and TTG were used by the *nad2* and *atp6* genes, respectively. The TAA was characterized by eight PCGs (*atp8*, *nad1*, *nad2*, *nad3*, *nad4L*, *nad5*, *nad6*, and *cytb*) as their stop codon, while TAG was used by the rest of the four genes, and the uncompleted stop codon “T” was used by *nad4*. The RSCU and amino acid usage in the PCGs are presented in Table 3. The most commonly occurring amino acids were arginine (Arg), leucine (Leu), valine (Val), serine (Ser), proline (Pro), threonine (Thr), alanine (Ala), and glycine (Gly), while tryptophan (Trp) was the least frequent (Figure 2). For the codon usage in the mitochondrial genome of *H. itama*, the three most commonly used were UUU, AUU, and AUA (Table 3).

### 3.3. Transfer RNAs and Ribosomal RNAs

All 22 tRNA genes were distributed over the mitogenome of *H. itama*, varying between 57 (*trnS1^Ser^*) and 70 (*trnY^Tyr^*) bp in length. The H-strand (+) of the mitogenome contained 14 tRNA genes, while the remaining 8 tRNA genes were situated on the L-strand (−) (Table 1 and Figure 3). The 22 tRNAs in the *H. itama* mitogenome sequence had a combined length of 1469 bp, accounting for 9.59% of the entire mitogenome (Table 1). These 22 tRNA genes exhibited characteristic cloverleaf secondary structures, except for *trnS1^Ser^* (UCU), which lacked a dihydrouridine (DHU) stem and loop (Figure 3).

Both *rrnS* and *rrnL* genes were located on the H-strand (+) of the mitochondrial genome. Together, they constituted 13.42% (2056 bp in length) of the entire *H. itama* mitogenome (Table 1). Positioned between *tRNA^Gln^* and *tRNA^Val^* genes, the *rrnS* gene is 761 bp long and has an A+T content of 75.82%. The *rrnL* gene, measuring 1295 bp with an A+T content of 77.52%, was situated between *tRNA^Val^* and *tRNA^Ser^* (Figure 1).

### 3.4. Intergenic Spacer and Overlapping Region

In the mitochondrial genome of *H. itama*, six overlapping regions between genes, totaling 8 bp in length, were identified. The lengths of these regions ranged from 1 bp, as observed between *tRNA^Trp^* and *tRNA^Tyr^*, *tRNA^Leu^* and *cox2*, *nad6* and *cytb*, and *cytb* and *tRNA^Ser^*, to 2 bp, found between *rrnS* and tRNA^Val^, as well as *rrnL* and *tRNA^Leu^*. The intergenic spacers in the *H. itama* mitogenome ranged from 1 to 100 bp in length, comprising 770 bp, and were distributed across 24 regions throughout the mitogenome (Table 1). The three largest intergenic spacers were located between *tRNA^Pro^* and *nad4l*, *tRNA^Arg^* and *tRNA^Gln^*, and *tRNA^Arg^* and *nad6*, with lengths of 100, 100, and 87 bp, respectively.

### 3.5. Phylogenomic Relationship

To gain a comprehensive understanding of genome-level evolution in *H. itama*, we conducted phylogenomic analyses using concatenated nucleotide sequences from 13 PCGs, 2 rRNA genes, and 22 tRNA genes, along with those from 22 representative hymenopteran insects with complete mitogenomes available in GenBank.

The phylogenetic analysis clearly grouped three major clades corresponding to the tribes Meliponini, Bombini, and Apini, with Colletes gigas (KM978210), *Euaspis polynesia* (MT909816), and *Megachile sculpturalis* (KT223644) serving as the outgroup. All nine meliponine-producing bees formed a monophyletic clade (Figure 4). *H. itama* from the present study was first grouped with two other species of *Tetragonula* (*T. iridipennis*: OQ139639; *T. pagdeni*: OK336459) and then combined with three other genera of stingless bees: *Lepidotrigona* (*L. terminate*: MN737481; *L. flavibasis*: MN747147), *Tetragonisca* (*T. angustula*: OR030859), and *Melipona* (*M. fusciculata*: MH680930, M*. scutellaris*: KP202303, and *M. bicolor*: AF466146).

### 3.6. Gene Rearrangement

The gene order of the *H. itama* mitochondrial genome closely resembled that of two *Tetragonula* species but differed from the gene arrangements found in six others previously reported stingless bee mitogenomes, which exhibited several gene rearrangements (Figure 5). The gene rearrangement events aligned with the phylogenetic tree derived from mitogenome sequences (Figure 4 and Figure 5). Additionally, variations in the position of *tRNA^Lys^* in the stingless bee mitogenome were observed. In *Heterotrigona* and *Tetragonula*, a *tRNA^Lys^*‒*tRNA^Ala^*‒*tRNA^Ile^*‒*tRNA^Met^* block was found, whereas in *Lepidotrigona* and *Melipona*, the block was altered to *tRNA^Met^*‒*tRNA^Lys^*‒*tRNA^Ala^*‒*tRNA^Ile^* and *tRNA^Ile^*‒*tRNA^Ala^*‒*tRNA^Lys^*‒*tRNA^Met^*, respectively (Figure 5).

In this study, we calculated the RS of each mitogenome by summing the RS values of all genes in the mitogenome. The RF of each gene was also computed using the mitogenome. Our results indicate that *H. itama* was a highly active group, comparable to stingless bees of the genus *Tetragonula*, and exhibited the highest RS among corbiculate bees (Figure 6). Among the individual genes, tRNA genes exhibited higher rearrangement frequencies, as observed in the PCGs (Figure 7). Among tRNA genes, *tRNA^Leu^* showed the highest rearrangement frequency. Apart from tRNA genes, PCGs and rRNA genes also underwent significant rearrangements in *H. itama*. The conserved *nad1*‒*tRNA^Leu^*‒*rrnL*‒*tRNA^Val^*‒*rrnS* block in Apini and Bombini was transposed into *Heterotrigona* and *Tetreagonula* (Figure 5). The conserved block of *nad1*‒*tRNALeu*‒*rrnL*‒*tRNAVal*‒*rrnS* found in Apini and Bombini was transposed in *Heterotrigona* and *Tetragonula* (Figure 5). Additionally, the *tRNA^Phe^*‒*nad5*‒*nad4*‒*nad4l* block in *Apis* species was transposed and inverted into *H. itama*. These observations indicate that *H. itama* has a relatively active mitochondrial genome.

## 4. Discussion

The mitochondrial genome of insects, including hymenopteran insects, is typically shown as a circular double-stranded molecule of approximately 14–19 kb in length. It generally contains 37 genes, comprising 13 PCGs, 2 ribosomal RNA (rRNA) genes, and 22 transfer RNA (tRNA) genes [24]. In this study, the total length of the complete mitochondrial genome of *H. itama* is 15,318 bp. This is consistent with the mitochondrial genome sizes of other stingless bees (*T. iridipennis*: 15,045 bp, *T. pagdeni*: 16,061 bp, *L. flavibasis*: 15,408 bp, *L. terminata*: 15,431 bp, and *M. bicolor*: 15,001 bp) and honeybees (*A. mellifera*: 16,336, *A. cerana*: 15,895 bp, *A. florea*: 15,993 bp, and *A. dorsata*: 15,892 bp). However, the mitochondrial genome of *H. itama* is slightly shorter than that of bumblebees, such as *B. terrestris* (17,232 bp) and *B. canariensis* (17,300 bp).

An apparent bias towards the nucleotide bases A and T (AT bias) was observed in the *H. itama* mitogenome sequence. Among the four nitrogenous bases, base T was the most abundant, whereas base G was the least abundant in the entire mitogenome. Furthermore, our findings confirm that the nucleotide composition of stingless bees has a high A+T content [14,24,45,46]. In general, both AT and GC skewness can be used to describe the base composition bias in nucleotide sequences [18]. Clary and Wolstenholme [47] reported that bias in nucleotide usage can be attributed to the active use of specific bases by DNA polymerases during mitochondrial DNA replication. Moreover, Xia [48] reported that subsequent transcription generates an AT bias in species relying on mitochondrial efficiency to sustain a high metabolic rate, as breaking A-T bonds requires less energy. This study revealed a similar pattern, with a negative AT skewness value (−0.083), suggesting a preference for base T over base A in the *H. itama* mitogenome.

As observed in other stingless bees, such as *M. bicolor* [49], *M. scutellaris* [50], *L. terminata* [51], *L. flavibasis* [14], *T. pagdeni* [46], and *T. iridipennis* [18], all 13 PCGs were allocated throughout the mitogenome of *H. itama*, with slight variations in size and position. The start codon (ATN) and the incomplete stop codon (T), which are commonly found in hymenopteran bees, were also observed in *H. itama*. No anomalous initiation codons were observed in any of the 13 PCGs of the *H. itama* mitogenome, unlike *L. flavibasis* (GTG start codon for *nad2*) [14] and *Apis cerana* (TTG start codon for *cox1*) [52]. In the universal genetic code, the TGA codon serves as a stop codon; however, in most mitochondrial genomes‒except those of higher plants–it encodes tryptophan [53]. The majority of stop codons use TAA for termination, though an incomplete stop codon (*nad4*) was identified in the *H. itama* mitogenome. This suggests that polyadenylation likely occurs post-transcription to finalize the termination codon [52]. Moreover, incomplete stop codons are commonly found in protein-coding genes (PCGs) of the Hymenoptera mitochondrial DNA sequenced so far [18,52,54,55,56]. Additionally, RSCU analysis indicated a preference for AT-rich codons in the *H. itama* mitogenome, primarily attributed to an AT mutational bias [46].

All 22 tRNA genes exhibited the typical cloverleaf structure, except for tRNASer, which lacked a dihydrouridine (DHU) arm and instead formed a simple loop (Figure 3). In most insect and metazoan mitochondria, t*RNA^Ser^* (UCU) has an unpaired DHU arm, *tRNA^Ser^* (UCU) generally has an unpaired DHU arm, and *tRNA^Ser^* (UGA) retains the standard cloverleaf structure [24,57]. The *H. itama tRNA^Ser^* (UCU) and *tRNA^Ser^* (UGA) structures confirmed this hypothesis. The putative secondary structures of tRNAs are similar to that of *T. iridipenis* [18], *L. flavibasis* [14], and *T. pagdeni* [46], indicating their similar function. The anticodons of tRNAs were similar to that of meliponine bees reported by various studies [14,18,46,49]. In the tRNAs of the *H. itama* mitogenome, the amino acid acceptor stem (15 bp) and the anticodon loop (7–9 bp) are highly conserved, resembling those observed in *T. pagdeni* [46], *L. flavibasis* [14], and *A. cerana* [52].

Mismatched base pairs in mitochondrial tRNAs have been documented in various hymenopteran insects, including *T. iridipenis* [18], *T. pagdeni* [46], *L. flavibasis* [14], *M. bicolor* [49], *A. cerana* [52], *A. mellifera* [45], *Bombus ignitus* [55], *Diadegma semiclausum* [56], and *Evania appendigaster* [58]. In this study, 10 mismatched base pairs were detected in the tRNAs of *H. itama*. Specifically, three tRNA genes (*tRNA^Cys^*, *tRNA^Gln^*, and *tRNA^Thr^*) exhibited single T-T mismatches in the acceptor stem, while seven others (*tRNA^Lys^*, *tRNA^Ala^*, tRNA^Leu^, *tRNA^Asp^*, *tRNA^Arg^*, *tRNA^Val^*, and *tRNA^Pro^*) had single G-T mismatches in the same region. Additionally, G-T mismatches were identified in the DHU arm of *tRNA^Gln^*, *tRNA^Pro^*, and *tRNA^Phe^* (Figure 3). The arrangement of two ribosomal RNA genes in the *H. itama* mitogenome was consistent with that of *T. iridipennis* and *T. pagdeni* but differed from those found in other stingless bees (*Lepidotrigona* and *Melipona* species), bumblebees, and honeybees. These variations resulted in their placement in distinct clades of the phylogenetic tree [14,18,49].

Phylogenetic analyses based on complete mitochondrial sequences are more reliable than those using partial or combined partial mitochondrial and nuclear gene sequences, as they produce topologies with strong support values [14,18]. In this study, the complete mitogenome of *H. itama*, along with those of 22 common hymenopteran species obtained from GenBank, was utilized for phylogenetic analysis. Our results indicate that *H. itama* and other stingless bee species are closely related with strong support (Figure 4), supporting taxonomic classification using morphological and molecular data. Three major clades were constructed within the Apidae family: Apini, Bombini, and Meliponini, which were highly supported in all analyses. These findings align with those of previous studies [18,46,58,59]. Our results strongly support a relationship among hymenopteran insects.

Most hymenopteran mitogenomes exhibit similar gene order [18,60,61]. Generally, gene rearrangements, including tandem duplication, inverse transposition, genetic transposition, and gene inversion, can cause the loss of gene function [24,26]. Consistent with that in other stingless bees, the absence of a conserved *tRNA^Asp^*–*tRNA^Lys^* block suggests that *tRNA^Lys^* movement may have occurred early in the progenitors of the stingless bee lineage, thus separating stingless bees from other hymenopterans (Apini and Bombini tribes) [46,52]. Our results also reveal that the mitogenome of *H. itama* shows a unique translocation that disrupted the relatively frequent intact *nad1*–*tRNA^Leu^*–*rrnL*–*tRNA^Val^*–*rrnS* block observed in hymenopteran mitogenomes. This finding implies that *H. itama* has experienced a high degree of mitogenomic plasticity. The mitogenomes of most invertebrate species largely retain their ancestral gene order. Nonetheless, natural selection may not entirely remove mitogenome rearrangements, reflecting the evolutionary dynamics and rates within a particular species or genus [62].

In the tropics and subtropics, numerous agricultural crops and wild plants depend mainly on insect pollinators for cross-pollination. Among insect pollinators, honeybees, stingless bees, and bumblebees, all belonging to the family Apidae, are the most important pollination vectors. The analysis of the *H. itama* mitogenome has enhanced our knowledge of hymenopteran mitochondrial genomes, offering new genetic markers for research in population genetics, systematics, phylogenomics, and biogeography of stingless bees.

## 5. Conclusions

The complete mitogenome of *H. itama* was sequenced using next-generation Illumina technology. The circular mitogenome spans 15,318 bp and comprises 37 sequence elements, including 13 protein-coding genes (PCGs), 2 rRNA genes, and 22 tRNA genes. Phylogenetic and gene order analyses revealed that *H. itama* is closely related to stingless bees of the genus *Tetragonula* but remains relatively distinct from the six previously reported stingless bee mitogenomes. The genetic data obtained from this study will aid future research in comparative genomics, molecular systematics, and population genetics of various hymenopteran bee species.

## Figures and Tables

**Figure 1 insects-16-00535-f001:**
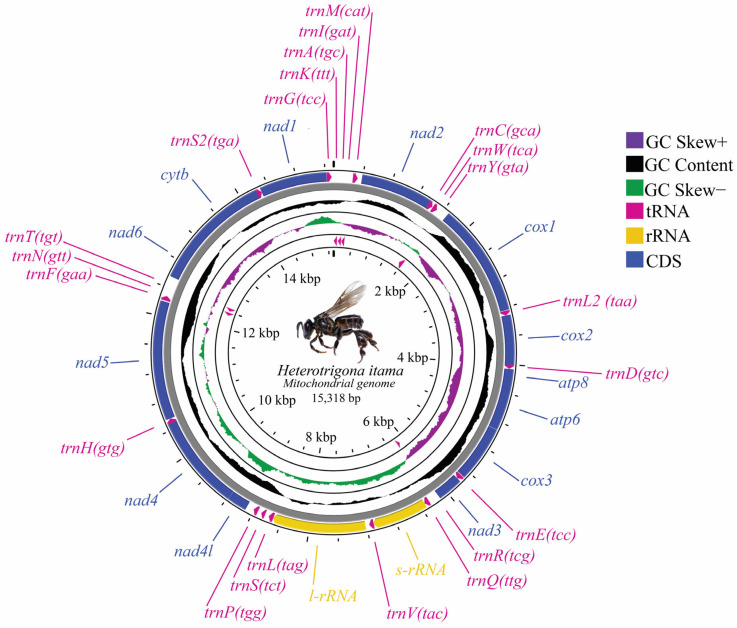
Mitogenome map of *Heterotrigona itama*. All 37 genes were distributed in different directions. The outer circle demonstrates the gene map, including 13 PCGs, 2 rRNAs, and 22 tRNAs. The tRNA genes are abbreviated by one-letter symbols according to the IUPAC-IUB single-letter amino acid codes. The second and third circles show the GC content and GC skew, respectively. GC content and GC skew are plotted as the deviation from the average value of the entire sequence.

**Figure 2 insects-16-00535-f002:**
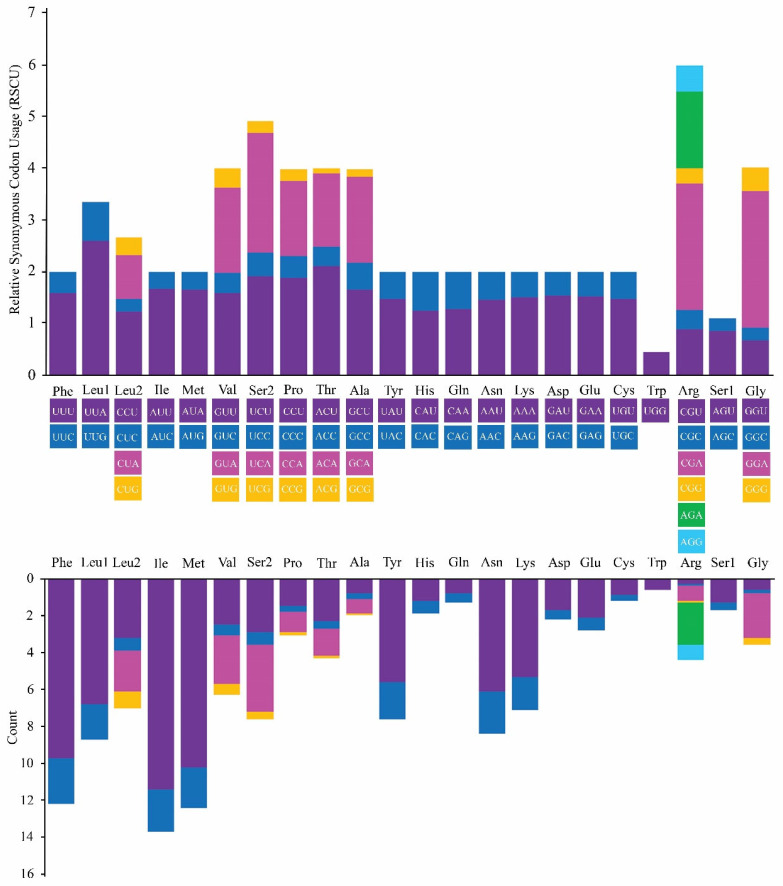
Relative synonymous codon usage (RSCU) in the *H. itama* mitogenome. The different colors in the column charter show that the codon families correspond to the amino acids below, and consistent colors represent the same codon families.

**Figure 3 insects-16-00535-f003:**
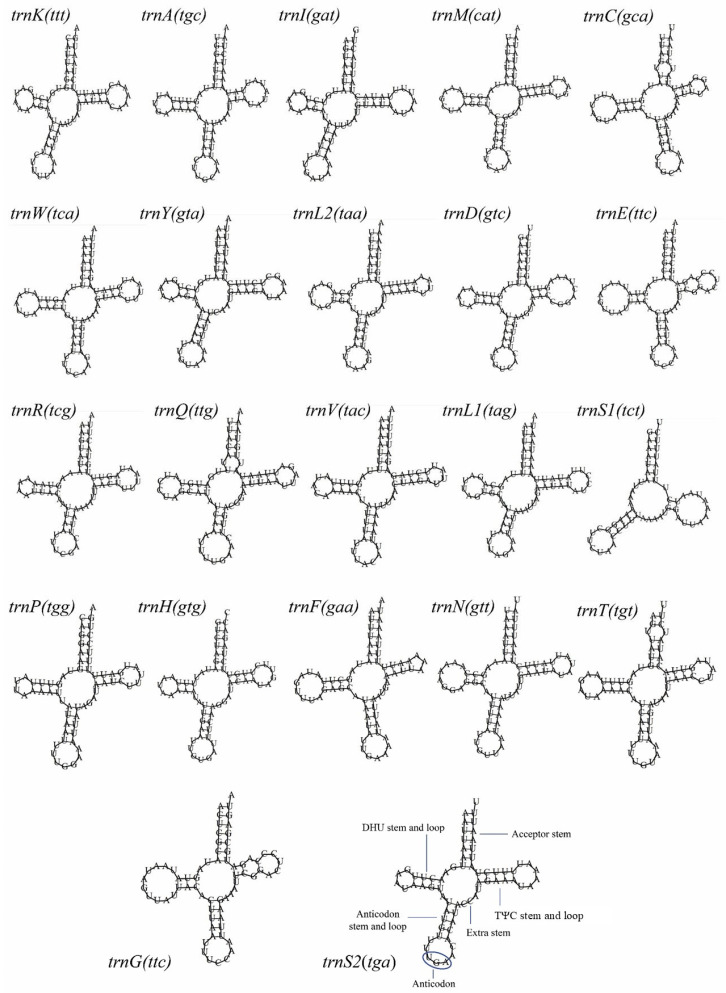
Predicted secondary cloverleaf structures for the 22 tRNA genes of the *H. itama* mitogenome.

**Figure 4 insects-16-00535-f004:**
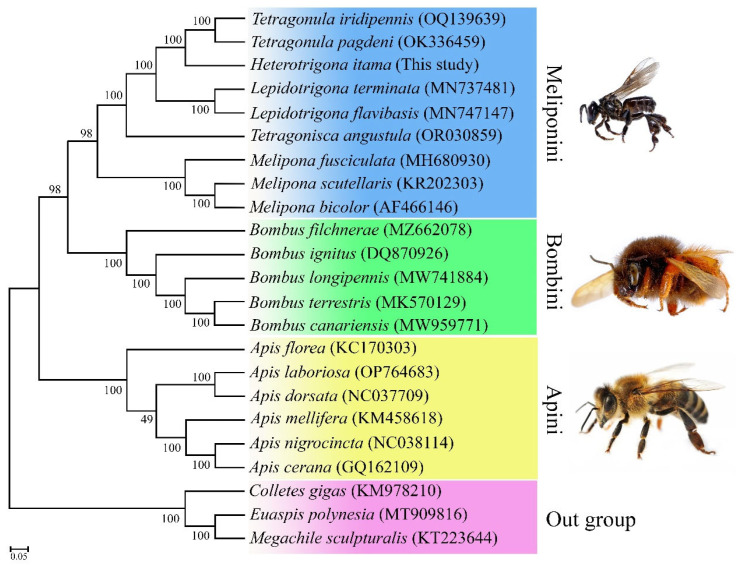
Phylogenetic tree showing the relationship between *H. itama* and 22 other Hymenopterans was constructed based on the concatenated sequences of the 37 mitochondrial coding genes of the whole mitogenome using maximum likelihood (ML) and Bayesian inference (BI) analysis. *Colletes gigas, Euaspis polynesia*, and *Megachile sculpturalis* were used as an outgroup. The GenBank accession numbers of each sequence are listed in the tree with their corresponding species names.

**Figure 5 insects-16-00535-f005:**
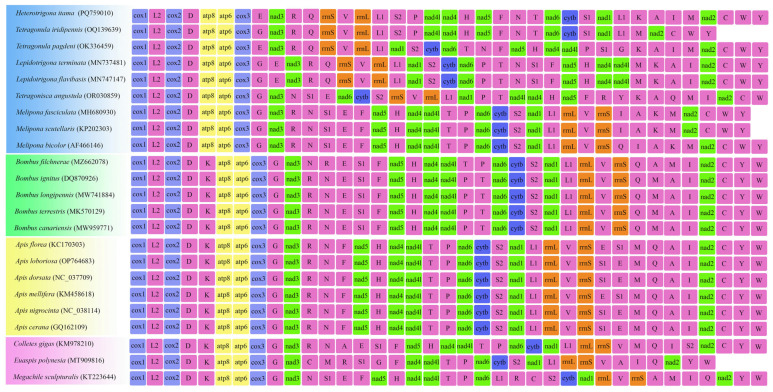
Gene order of each mitochondrial genome. The same color shows the sequence of similar genes. The uppercase alphabet letters are the single-letter abbreviation of each amino acid.

**Figure 6 insects-16-00535-f006:**
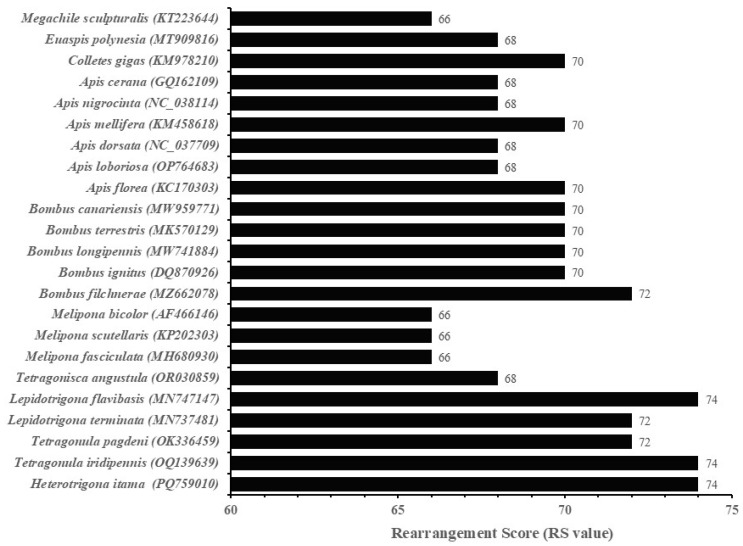
Rearrangement score (RS value) of each mitogenome.

**Figure 7 insects-16-00535-f007:**
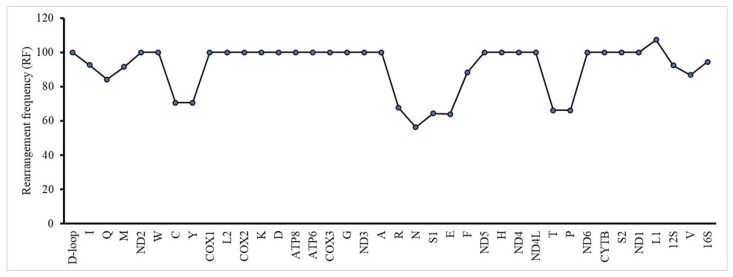
Rearrangement frequency (RF) of each single mitochondrial gene in all the analyzed mitogenomes.

**Table 1 insects-16-00535-t001:** Annotation of the *Heterotrigona itama* mitochondrial genome. * TAA stop codon was completed by the addition of 3′ A residues in polycistronic transcription cleavage and polyadenylation processes.

Locus	Full Name and Function	Position		Length (bp)	Strand	Intergenic Spacer	Codon		Anti- Codon
		Start	End				Start	Stop	
*tRNA-Lys (ttt)*	Transfer RNA for Lysine	1	69	69	−	22			TTT
*tRNA-Ala (tgc)*	Transfer RNA for Alanine	92	156	65	−	8			TGC
*tRNA-Ile (gat)*	Transfer RNA for Isoleucine	165	231	67	−	35			GAT
*tRNA-Met (cat)*	Transfer RNA for Methionine	267	335	69	+	56			CAT
*nad2*	NADH dehydrogenase subunit 2	392	1383	992	+	0	ATC	TAA	
*tRNA-Cys (gca)*	Transfer RNA for Cysteine	1384	1448	65	+	8			GCA
*tRNA-Trp (tca)*	Transfer RNA for Tryptophan	1457	1524	68	+	−1			TCA
*tRNA-Tyr (gta)*	Transfer RNA for Tyrosine	1524	1611	88	−	69			GTA
*cox1*	Cytochrome c oxidase subunit I	1681	3240	1560	+	6	ATT	TAG	
*tRNA-Leu (taa)*	Transfer RNA for Leucine	3247	3312	66	+	0			TAA
*cox2*	Cytochrome c oxidase subunit II	3313	3999	687	+	−1	ATT	TAG	
*tRNA-Asp (gtc)*	Transfer RNA for Aspartic acid	3999	4056	58	+	2			GTC
*atp8*	ATP synthase F0 subunit 8	4059	4226	168	+	17	ATT	TAA	
*atp6*	ATP synthase F0 subunit 6	4244	4909	666	+	4	TTG	TAG	
*cox3*	Cytochrome c oxidase subunit III	4914	5693	780	+	8	ATG	TAG	
*tRNA-Glu(ttc)*	Transfer RNA for Glutamic acid	5702	5769	68	+	0			TTC
*nad3*	NADH dehydrogenase subunit 3	5770	6123	354	+	0	ATA	TAA	
*tRNA-Arg(tcg)*	Transfer RNA for Arginine	6124	6185	62	−	87			TCG
*tRNA-Gln(ttg)*	Transfer RNA for Glutamine	6273	6340	68	+	1			TTG
*s-rRNA*	12S ribosomal RNA	6342	7102	761	+	−2			
*tRNA-Val(tac)*	Transfer RNA for Valine	7101	7165	65	+	53			TAC
*l-rRNA*	16S ribosomal RNA	7219	8513	1295	+	−2			
*tRNA-Leu(tag)*	Transfer RNA for Leucine	8512	8579	68	+	55			TAG
*tRNA-Ser(tct)*	Transfer RNA for Serine	8635	8691	57	+	52			TCT
*tRNA-Pro(tgg)*	Transfer RNA for Proline	8744	8807	64	+	100			TGG
*nad4l*	NADH dehydrogenase subunit 4L	8908	9180	273	+	2	ATT	TAA	
*nad4*	NADH dehydrogenase subunit 4	9183	10485	1303	+	0	ATG	T (* AA)	
*tRNA-His(gtg)*	Transfer RNA for Histidine	10486	10551	66	+	0			GTG
*nad5*	NADH dehydrogenase subunit 5	10552	12204	1653	+	16	ATT	TAA	
*tRNA-Phe(gaa)*	Transfer RNA for Phenylalanine	12221	12286	66	+	20			GAA
*tRNA-Asn(gtt)*	Transfer RNA for Asparagine	12307	12375	69	−	30			GTT
*tRNA-Thr(tgt)*	Transfer RNA for Threonine	12406	12471	66	−	87			TGT
*nad6*	NADH dehydrogenase subunit 6	12559	13077	519	+	−1	ATA	TAA	
*cytb*	Cytochrome b	13077	14225	1149	+	−1	ATG	TAA	
*tRNA-Ser(tga)*	Transfer RNA for Serine	14225	14291	67	+	0			TGA
*nad1*	NADH dehydrogenase subunit 1	14292	15218	927	−	3	ATT	TAA	
*tRNA-Gly (tcc)*	Transfer RNA for Glycine	15222	15289	68	−	29			TCC

**Table 2 insects-16-00535-t002:** Nucleotide composition and GC/AT skews in the 13 protein-coding genes (PCGs)and 2 rRNAs of the *Heterotrigona itama* mitochondrial genome.

Gene	Length	T%	C%	A%	G%	AT%	GC%	GC Skew	AT Skew
*nad2*	996	40.7	15.9	31.2	12.2	71.9	28.1	−0.131	−0.132
*cox1*	1575	39.34	18.16	27.92	14.58	67.26	32.74	−0.109	−0.169
*cox2*	676	37.66	17.7	30.94	13.7	68.6	31.4	−0.127	−0.098
*atp8*	168	39.78	15.86	36.72	7.64	76.5	23.5	−0.349	−0.040
*atp6*	666	39.76	21.12	25.76	13.36	65.52	34.48	−0.225	−0.21368
*cox3*	780	36.68	19.76	30.38	13.18	67.06	32.94	−0.199	−0.093
*nad3*	354	39.4	21.2	25.82	13.58	65.22	34.78	−0.219	−0.208
*nad4l*	300	48.4	5.7	36.04	9.86	84.44	15.56	0.267	−0.146
*nad4*	1303	46.16	8.16	36.38	9.3	82.54	17.46	0.065	−0.118
*nad5*	1653	44.56	7.64	39.86	7.94	84.42	15.58	0.019	−0.056
*nad6*	501	46.5	8.6	38.74	6.16	85.24	14.76	−0.165	−0.091
*cytb*	1149	45	12.82	32.62	9.56	77.62	22.38	−0.146	−0.159
*nad1*	924	45.68	8.44	35.86	10.02	81.54	18.46	0.086	−0.120
*rrnS*	761	35.26	8.98	40.56	15.2	75.82	24.18	0.257	0.070
*rrnL*	1316	38.2	9.58	39.32	12.9	77.52	22.48	0.148	0.014
Overall Mitogenome	15318	41.42	12.22	35.06	11.30	75.413	24.587	−0.039	−0.083

**Table 3 insects-16-00535-t003:** Summary of codon usage and relative synonymous codon usage (RSCU) pattern of the 13 PCGs from the *Heterotrigona itama* mitochondrial genome. Average codons = 124.

Aa	Codon	N	RSCU	Aa	Codon	N	RSCU	Aa	Codon	N	RSCU	Aa	Codon	N	RSCU
Phe	UUU(F)	9.7	1.59	Ser	UCU(S)	2.9	1.92	Tyr	UAU(Y)	5.6	1.47	Cys	UGU(C)	0.9	1.48
	UUC(F)	2.5	0.41		UCC(S)	0.7	0.45		UAC(Y)	2.0	0.53		UGC(C)	0.3	0.52
Leu	UUA(L)	6.8	2.60		UCA(S)	3.6	2.32	End	UAA (*)	2.1	1.55	End	UGA (*)	2.1	1.56
	UUG(L)	1.9	0.74		UCG(S)	0.4	0.23		UAG (*)	0.6	0.45	Trp	UGG(W)	0.6	0.44
	CUU(L)	3.2	1.22	Pro	CCU(P)	1.5	1.88	His	CAU(H)	1.2	1.25	Arg	CGU(R)	0.3	0.89
	CUC(L)	0.7	0.26		CCC(P)	0.3	0.43		CAC(H)	0.7	0.75		CGC(R)	0.1	0.37
	CUA(L)	2.2	0.84		CCA(P)	1.1	1.45	Gln	CAA(Q)	0.8	1.28		CGA(R)	0.8	2.44
	CUG(L)	0.9	0.34		CCG(P)	0.2	0.23		CAG(Q)	0.5	0.72		CGG(R)	0.1	0.30
Ile	AUU(I)	11.4	1.66	Thr	ACU(T)	2.3	2.11	Asn	AAU(N)	6.1	1.46	Ser	AGU(S)	1.3	0.85
	AUC(I)	2.3	0.34		ACC(T)	0.4	0.38		AAC(N)	2.3	0.54		AGC(S)	0.4	0.25
Met	AUA(M)	10.2	1.65		ACA(T)	1.5	1.41	Lys	AAA(K)	5.3	1.50	Arg	AGA(R)	2.3	1.48
	AUG(M)	2.2	0.35		ACG(T)	0.1	0.1		AAG(K)	1.8	0.50		AGG(R)	0.8	0.50
Val	GUU(V)	2.5	1.59	Ala	GCU(A)	0.8	1.65	Asp	GAU(D)	1.7	1.54	Gly	GGU(G)	0.6	0.68
	GUC(V)	0.6	0.39		GCC(A)	0.3	0.53		GAC(D)	0.5	0.46		GGC(G)	0.2	0.23
	GUA(V)	2.6	1.64		GCA(A)	0.8	1.65	Glu	GAA(E)	2.1	1.52		GGA(G)	2.4	2.65
	GUG(V)	0.6	0.38		GCG(A)	0.1	0.16		GAG(E)	0.7	0.48		GGG(G)	0.4	0.45

Aa = Amino acid; N = frequency of each codon; RSCU = relative synonymous codon usage. Asterisks (*) indicate termination codon.

## Data Availability

All relevant data are within the paper.

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
