# Peer review of "Complete Mitochondrial Genome Characterization and Phylogenomics of the Stingless Bee, Heterotrigona itama (Apidae: Meliponini)"

_insects, 2025, doi:10.3390/insects16050535_

Round 1

Reviewer 1 Report

Comments and Suggestions for Authors

The manuscript “Complete mitochondrial genome characterization and phylogenomics of stingless bee, Heterotrigona itama (Apidae: Meliponini)". I consider that the experiments are well designed, and the results are processed by statistical analysis. The detailed comments are as follows:

  1. The paper is missing Simple SummaryAn , Example could be found at https://www.mdpi.com/2075-4450/11/8/508.
  2. Abstracts have too many words, insects require a total of about 200 words maximum.
  3. Line37, “H. itama. The ecologically”
  4. Line 44, among of the most
  5. Line 52, “for” should be “in”
  6. Line 54, “being” should be “is”
  7. Line 59, keeping of itam.
  8. Line 60, “the” should be “The”.
  9. Line 62, allocated “an at-risk”.
  10. Line 84, consists “of”.
  11. Line 103, the reference style of “Samsudin et al. (2018)” and “Trianto et al. (2023)” was not right.
  12. Line 113, What is the number of samples used for DNA extraction? larvae? pupae or adult?
  13. Line 119, “isolated” should be “isolate”
  14. Tables 1 and 2 should be three-line tables.
  15. Line 248, H. itama should be in italics
  16. The formatting of the references does not meet the requirements of the journal and needs to be carefully revised.

Author Response

Respond to reviewer 1

Comments and Suggestions for Authors

The manuscript “Complete mitochondrial genome characterization and phylogenomics of stingless bee, Heterotrigona itama (Apidae: Meliponini)". I consider that the experiments are well designed, and the results are processed by statistical analysis. The detailed comments are as follows:

Comment 1: The paper is missing Simple Summary, Example could be found at https://www.mdpi.com/2075-4450/11/8/508.

Author responds: The simple summary was added.

Comment  2: Abstracts have too many words, insects require a total of about 200 words maximum.

Author responds:

Comment  3: Line37, “H. itama. The ecologically”

Author responds: It has been revised in the new abstract version.

Comment  4: Line 44, among of the most

Author responds: Agree, it has been changed.

Comment  5: Line 52, “for” should be “in”

Author responds: Agree, it has been changed.

Comment 6: Line 54, “being” should be “is”

Author responds: Agree, it has been changed.

Comment 7: Line 59, beekeeping of H. itama.

Author responds: Agree, it has been changed.

Comment 8: Line 60, “the” should be “The”.

Author responds: Agree, it has been changed.

Comment  9: Line 62, allocated “an at-risk”.

Author responds: Agree, it has been changed.

Comment 10: Line 84, consists “of”.

Author responds: Agree, it has been changed.

Comment 11: Line 103, the reference style of “Samsudin et al. (2018)” and “Trianto et al. (2023)” was not right.

Author responds: It has been revised.

Comment 12: Line 113, What is the number of samples used for DNA extraction? larvae? pupae or adult?

Author responds: The more detail of bee sample has been added in Lines 82-84 and 101-102 in revised version.

Comment 13: Line 119, “isolated” should be “isolate”

Author responds: Agree, it has been changed.

Comment 14: Tables 1 and 2 should be three-line tables.

Author responds: Agree, the tables have been changed.

Comment 15: Line 248, H. itama should be in italics

Author responds: It has been corrected.

Comment 16: The formatting of the references does not meet the requirements of the journal and needs to be carefully revised.

Author responds: All references both in-text cites and list were amended.

Reviewer 2 Report

Comments and Suggestions for Authors

Dear authors,

The first complete mitogenome of one of the most significant stingless bees in terms of honey production is reported in the manuscript "Complete mitochondrial genome characterisation and phylogenomics of the stingless bee, Heterotrigona itama (Apidae: Meliponini)”, which has greatly advanced our knowledge of stingless bee genetics. The study offers crucial information about the genetic makeup and evolution of this economically and ecologically important species. Complete mitochondrial data will help future efforts in species identification, population genetics, and conservation, especially since H. itama is becoming more and more important in meliponiculture and as a pollinator in tropical ecosystems.

The work is well-structured and organized, as well as the figures being attractive. The English is good but can be improved, and only a few problems were detected and are stated below:

Title:

Line 1-4: “the” is missing before “stingless bee”

Abstract:

Line: 14: “and” is missing before "extensively”

Line 16: “H. itama” should be in italic

Introduction:

Line 49: replace “ease of keeping” with “ease of beekeeping”. Replace “keeping” with beekeeping all over the text.

Line 60: Capital letter at the beginning of the sentence is missed.

Line 75: The word “Owing” is repeated closely in the text. Find another way to write.

Line 89: delete “of the mitogenome”

Materials and Methods:

Line 120: delete “.” before “according”

Line 148-149: Delete the second time that appear corbiculate bees.

Line 244-246: This sentence is repeated, please delete it.

Line 248: H. itama in italic

Line 250: Delete “H. itama”

Question: Why did you not use the reference mitogenome for A. mellifera in phylogenetic analysis?

Discussion:

No comments

Conclusion:

No comments

Author Response

Respond to Comment 2

Comments and Suggestions for Authors

Dear authors,

The first complete mitogenome of one of the most significant stingless bees in terms of honey production is reported in the manuscript "Complete mitochondrial genome characterisation and phylogenomics of the stingless bee, Heterotrigona itama (Apidae: Meliponini)”, which has greatly advanced our knowledge of stingless bee genetics. The study offers crucial information about the genetic makeup and evolution of this economically and ecologically important species. Complete mitochondrial data will help future efforts in species identification, population genetics, and conservation, especially since H. itama is becoming more and more important in meliponiculture and as a pollinator in tropical ecosystems.

The work is well-structured and organized, as well as the figures being attractive. The English is good but can be improved, and only a few problems were detected and are stated below:

Title:

Comment 1: Line 1-4: “the” is missing before “stingless bee”

Author responds: Agree, “the” has been added in the Title.

Abstract:

Comment 2: Line: 14: “and” is missing before "extensively”

Author responds: It has been revised in the new abstract version.

Comment 3: Line 16: “H. itama” should be in italic

Author responds: It has been revised in the new abstract version.

Introduction:

Comment 4: Line 49: replace “ease of keeping” with “ease of beekeeping”. Replace “keeping” with beekeeping all over the text.

Author responds: The word “beekeeping” has been replaced.

Comment 5: Line 60: Capital letter at the beginning of the sentence is missed.

Author responds: It has been corrected.

Comment 6: Line 75: The word “Owing” is repeated closely in the text. Find another way to write.

Author responds: Agree, the word “Due to” has been replaced.

Comment 7: Line 89: delete “of the mitogenome”

Author responds: Agree, it has been deleted.

Materials and Methods:

Comment 8: Line 120: delete “.” before “according”

Author responds: “.” has been deleted.

Comment 9: Line 148-149: Delete the second time that appear corbiculate bees.

Author responds: the word “of corbiculate bees” at the end of sentence (Line 136-137 of revised version) has been deleted.

Comment 10: Line 244-246: This sentence is repeated, please delete it.

Author responds: The repeated sentence has been deleted.

Comment 11: Line 248: H. itama in italic

Author responds: It has been corrected.

Comment 12: Line 250: Delete “H. itama”

Author responds: It has been deleted (Line 235 of revised version).

Comment 13: Question: Why did you not use the reference mitogenome for A. mellifera in phylogenetic analysis?

Author responds: In this study, we used the most closely related group as a re reference mitogenome, which are three species of stingless bee [Tetragonula iridipenis (OQ139639), Tetragonula pagdeni (OK336459.1), and Lepidotrigona flavibasis (MN747147)] native to the same locality with H. itama. While, A. mellifera, introduced to Thailand and other Southeast Asian country, shows more genetic distance.

Discussion:

Comment 14: No comments

Author responds: -

Conclusion:

Comment 15: No comments

Author responds: -
